# Dysfunction of Biliary Sphincter of Oddi—Clinical, Diagnostic and Treatment Challenges

**DOI:** 10.3390/jcm12144802

**Published:** 2023-07-20

**Authors:** Marina Kegnæs, Srdan Novovic, Daniel Mønsted Shabanzadeh

**Affiliations:** 1Pancreatitis Centre East, Gastrounit, Copenhagen University Hospital—Amager and Hvidovre, Hvidovre, 1172 Copenhagen, Denmark; m.kegnaes@gmail.com; 2Department of Clinical Medicine, University of Copenhagen, 1172 Copenhagen, Denmark; 3Department of Surgery, North Zealand University Hospital, Dyrehavevej 29, 3400 Hillerød, Denmark; daniel.moensted.shabanzadeh.01@regionh.dk

**Keywords:** cholecystectomy, post-cholecystectomy pain, sphincter of Oddi dysfunction (SOD), sphincterotomy, cholescintigraphy

## Abstract

Biliary Sphincter of Oddi dysfunction (SOD) is one of the main causes of post-cholecystectomy pain. In this review, we aimed to provide an update on the current knowledge on biliary SOD, with an emphasis on diagnostics and therapy. Overall, current but scarce data support biliary sphincterotomy for patients with type 1 and 2 SOD, but not for type 3. However, sphincterotomy is associated with post-treatment pancreatitis rates of from 10% to 15%, thus calling for improved diagnostics, patient selection and treatment modalities for SOD. The role of pharmacologic therapy for patients with SOD is poorly explored and only two randomized controlled trials are available. Currently, studies comparing treatment outcomes are few. There is an unmet need for randomized sham/placebo-controlled clinical trials related to both pharmacological and non-pharmacological treatments of SOD.

## 1. Introduction

Post-cholecystectomy pain refers to the subsequent development of recurrent episodes of abdominal pain in patients who have had cholecystectomy [1,2]. This pain occurs in 10–20% of post-cholecystectomy patients [3]. The etiologies of persistent and incident symptoms after cholecystectomy and the underlying mechanisms could be different [4]. Common biliary causes include retained or recurrent common duct stones, fibrosis due to chronic inflammation of the bile duct, or, less commonly, an inflamed cystic duct remnant [5,6]. Sphincter of Oddi dysfunction (SOD) is another important cause of post-cholecystectomy pain that is defined as an abnormality of either the biliary and/or pancreatic sphincter causing intermittent or fixed obstruction to the flow of bile or pancreatic juice, associated with episodes of biliary-type pain, recurrent pancreatitis, elevated liver enzymes, or ductal dilatation [6,7,8,9,10]. Among those 10–20% of patients undergoing cholecystectomy who experience post-cholecystectomy biliary pain, 9–51% meet the diagnostic criteria for SOD [3,11]. Overall, patients with previous cholecystectomy are predisposed to SOD [11,12] and about 1.5% of patients develop SOD after cholecystectomy [13,14,15].

The existing literature on the topic, as well as the clinical evidence, are both limited. Moreover, the ultimate treatment with endoscopic retrograde cholangiopancreatography (ERCP) and sphincterotomy is associated with post-ERCP pancreatitis rates ranging from up to 10% to 15% [2,16,17,18], with possible long-term carcinogenic effects [19]. Furthermore, modern trends in healthcare require non-invasive methods for diagnosis and treatment in order to further reduce the risk of adverse events and complications. Therefore, there is an unmet need for randomized controlled clinical studies evaluating non-invasive diagnostics and interventions for the treatment of SOD, calling for improved diagnostics and treatment modalities [20]. In this review, we aimed to provide an update on the current knowledge on biliary SOD, with an emphasis on diagnostics and therapy [2,16,17,18].

## 2. Anatomy and Pathophysiology

The Sphincter of Oddi (SO) is regulating the flow of biliary and pancreatic secretion into the duodenum [14]. The SO is a complex muscular structure that surrounds the intraduodenal segment of the common bile duct, the pancreatic duct and major duodenal papilla, and functions independently of the duodenal musculature [21,22]. Italian anatomist, Ruggero Oddi, initially described the SO in 1887 [14]. A clinical syndrome that is defined as an abnormality of either the biliary and/or pancreatic sphincter related to the intermittent or fixed obstruction to the flow of bile or pancreatic juice is defined as SOD [8,9]. The physiology of the SO is quite complex [23]. SOD includes both mechanical and functional component. SO stenosis is an anatomic (mechanical) abnormality related to the narrowing of the SO or its obstruction, that could result from any process causing inflammation or scarring (pancreatitis, gallstone passage, infection, malignancy) [21]. SO dyskinesia is related to a functional disturbance in muscular tone control causing an intermittent biliary obstruction.

Basal pressure of the SO is 10 mmHg in humans. Superimposed anterograde phasic contractions, initiated at the junction of the common bile duct (CBD) and the SO and progressing into the duodenum, occur in response to physiologic and exogenous stimuli and result in the evacuation of contents already present within the SO into the duodenum. During contraction, no additional flow from the CBD into the SO occurs. The SO then relaxes, allowing for the passive refilling of bile into the SO segment. Once filled, another wave of phasic contractions begins. When basal pressure increases, resistance to flow increases, resulting in gallbladder filling and the prevention of flow into the duodenum. When basal pressure decreases below CBD and pancreatic duct (PD) pressures, flow into the duodenum occurs [14].

The motility of the SO during the digestive period involves both neural and hormonal input [23,24,25,26,27]. In this period, there are gallbladder contractions, stimulation of pancreatic secretion, and SO relaxation, leading to high rates of bile and pancreatic secretion into the duodenum. Cholecystokinin (CCK) is the most important hormone involved in SO motility. CCK decreases SO basal pressures and inhibits phasic contractions, thereby promoting anterograde flow. Enteroendocrine cells of pancreas secrete CCK in response to a meal, leading to gallbladder contraction, the relaxation of SO and the secretion of pancreatic enzymes, via both direct action on CCK receptors and indirect action through cholinergic neurons [14]. After cholecystectomy, the normal biliary tree is transformed into a single-outlet system in which intrahepatic ductal bile flows only through the bile duct into the duodenum [22]. Moreover, the biliary system no longer has a pressure-release mechanism, formerly served by gallbladder. If the SO resists flow, increased pressure produces pain [1] and may lead to biliary stasis and ductal dilation [22]. 

Thus, gallbladder function appears to play a critical role in SO mechanics; therefore, patients, after cholecystectomy, are more likely to develop SOD. It is also shown that, in patients with an intact gallbladder, CCK inhibits the SO phasic wave activity, but 6 months after cholecystectomy, CCK fails to inhibit this activity [11,28]. Furthermore, it is demonstrated that patients with post-cholecystectomy pain had elevated basal SO pressure, retrograde phasic wave contraction, and an increase in phasic wave frequency greater than seven contractions per minute. However, it is still unclear whether post cholecystectomy patients develop SOD due to elevated basal pressure, altered motility of the SO, or both [11,14]. Moreover, female gender, hypothyroidism, irritable bowel syndrome, prior pancreatitis and some medical drugs such as opiates are other potential contributing factors for SOD [14].

## 3. Clinical Manifestations, Classifications, and Laboratory Findings

Episodes of biliary pain are one of the most common manifestations of SOD, as well as recurrent pancreatitis, elevated liver enzymes, or ductal dilatation. Symptoms of biliary pain ascribed to gallstones have traditionally been termed “biliary colic” and were originally defined by a sudden debut of intense and agonizing pain, localized in the right hypochondrium or epigastrium, with projection to the shoulder, and a duration ranging from hours to weeks. Later, more simple definitions of biliary pain were formulated in gallstone screening studies including abdominal pain during the last five years, with a duration of more than 30 min, and with localization in the right hypochondrium and/or epigastrium [7]. Biliary SOD is characterized by intermittent or episodic right upper quadrant pain that might not necessarily be postprandial and is often accompanied by nausea and vomiting. Based on these criteria, a few classifications of SOD have been developed over time. 

As a structural abnormality, biliary SOD is classified according to Milwaukee classification into types 1, 2 and 3 based on clinical presentation as well as laboratory and/or imaging abnormalities; see Table 1 [8,11,12,14,29,30,31,32,33]. Diagnosis of type 3 SOD is challenging due to the lack of significant abnormalities on laboratory testing or diagnostic imaging.

As a functional disorder, SOD is classified according to Rome IV criteria that are linked closer to clinical practice, based mainly upon expert consensus [14,21,34].

Rome IV criteria for biliary SOD are presented in Table 2 [21,34,35]. At present, patients with post-cholecystectomy pain and the SOD are mostly diagnosed by Rome IV criteria. The symptoms, however, are also common for other disorders that are more frequent than SOD [13]; therefore, differential diagnostics should always be considered.

## 4. Diagnostic Challenges

In patients with previous cholecystectomy, the Rome IV consensus conference statement suggests that the reasonable approach towards the diagnosis of biliary SOD is to start with liver and pancreatic biochemical tests, followed by an upper endoscopy and abdominal imaging such as transabdominal ultrasound (US), computed tomography (CT) scan, magnetic resonance cholangiopancreatography (MRCP) or endoscopic ultrasound (EUS).

Previously, an elevation of basal sphincter pressure detected on endoscopic Sphincter of Oddi manometry (SOM) was the gold standard for establishing SOD diagnosis [2,5,36,37]. SOM is performed during ERCP and is the only investigation that can directly assess SO motor activity [38,39]. However, this method has mostly been abandoned, as it is an invasive procedure with a high risk of severe complications, such as post-ERCP pancreatitis rates of up to 30% [8,10,14,21,34,37,38,40,41,42,43]. Additionally, SOM is not a widely available technique due to the need for expertise to correctly perform and interpret the obtained manometric results [39,44].

The use of less invasive diagnostic methods for SOD is needed in order to replace the manometry measurements. Several new diagnostic modalities, such as functional Magnetic Resonance Imaging (functional MRI), Optical Coherence Tomography (OCT) and Functional Lumen Imaging Probe (FLIP) technique, have emerged in recent years [10,37,44,45]; see Table 3.

Functional MRI is a class of imaging methods that is quite popular due to its wide availability, non-invasive nature, relatively low cost and good resolution. The visualization of the biliary tract is possible using gadolinium-based contrast agents that are taken up by normal hepatocytes and partially excreted in the biliary system [46]. Functional MRI is similar to hepatobiliary scintigraphy, but with a higher resolution, and the hypothesis that SOD is caused by the delayed emptying of a biliary excreted contrast agent to the duodenum is investigated [37]. 

Optical coherence tomography permits the high-resolution, real-time imaging of the SO microstructure by a probe inserted into the common bile duct through an ERCP catheter [45]. OCT uses low-power infrared light in order to highlight the details of the microstructure of the gastrointestinal wall layer in real time, and it allows for a much higher resolution with a much lower depth penetration compared to US. The intermediate layer of the papillary region in patients with type 1 SOD was 2.3 times thicker than that in control patients and its infrared light back-scattering showed the hyper-reflectivity of the fibromuscular layer of SO as compared to controls [37,45]. The FLIP technique is used to analyze the sphincter profile and motility patterns of the sphincter during ERCP. Botox injections temporarily interfere with SO motility, causing its relaxation and, in some cases, pain relief [10]. At present, no strong evidence indicates that the less invasive methods have a better diagnostic accuracy compared to SOM [37]. 

Hepatobiliary scintigraphy (cholescintigraphy) is a promising diagnostic imaging tool that has found broad clinical application. It is quantitative, reliable, and non-invasive method for diagnosing SOD in patients who have had cholecystectomy, although the clinical value of this modality is not well established. Hepatobiliary scintigraphy involves the intravenous injection of ^99m^Tc-radioisotope compounds that are bound to serum albumin in the blood stream, which are dissociated from albumin in the hepatic presinusoidal space and extracted by hepatocytes by receptor-mediated endocytosis, similar to bile salts, free fatty acids and bilirubin. Afterwards, they follow the same metabolic pathway as bilirubin, except that they are secreted into biliary caliculi unchanged, without undergoing conjugation [1,21]. The time–activity curves for the radionucleotide excretion throughout the hepatobiliary system are measured [1,13,23,38,44,47]. This technique is used to assess the rate of bile flow into the duodenum. The literature regarding cholescintigraphy is relatively limited and quite challenging to interpret due to variations in methodology, clinical indications, categories of patients, time to peak activity, hepatic clearance at predefined time intervals, and a lack of consensus on its diagnostic use [1,10,21,22,23,38,44,47]. 

Compared to SOM, cholescintigraphy is increasingly used as a non-invasive method for diagnosing biliary SOD. In a previous, small study [38] quantitative scintigraphy was compared to SOM, and the results indicated that although scintigraphy was less sensitive than SOM, it successfully predicted the clinical outcome of biliary SOD after sphincterotomy in 93% of patients compared with 57% with SOM. 

## 5. Treatment Options

### 5.1. Non-Pharmacologic Treatment

The management of SOD most often involves non-pharmacologic treatments such as ERCP and sphincterotomy [10,14], and the correlation between type of SOD and clinical outcome after biliary sphincterotomy is continuously being explored [8]. Endoscopic sphincterotomy is the most commonly used non-pharmacologic treatment for SOD for patients with type 1 and 2 SOD [10]. 

Two randomized controlled trials [2,17], including a sphincterotomy group and a control group receiving sham intervention, showed that patients with abnormally elevated basal pressure showed over 90% long-term relief in symptoms related to biliary-type pain when undergoing sphincterotomy of the biliary SO [48].

Another study included 47 patients with biliary-type pain after cholecystectomy and clinical characteristics suggesting biliary obstruction [17]. All patients were subjected to SOM. After sphincterotomy, 10 of the 11 patients who had elevated basal pressures (greater than 40 mmHg) reported a statistically significant improvement in symptoms in comparison with 3 of 12 patients in the sham-treated group. During the four-year follow-up, an improvement in symptoms was demonstrated in 17 of 18 patients with elevated pressures who underwent sphincterotomy. Moreover, outcomes for sphincterotomy were not found to be significantly different compared to the sham procedure in patients who had normal manometric findings [17]. Similar outcomes were found in the randomized controlled trial performed by Toouli et al [2]. In this trial, 81 patients with SOD were included, and their manometric results were categorized as normal, SO stenosis or SO dyskinesia [2]. An assessment of pain status in patients at 24 months showed that the only group of patients that benefited from sphincterotomy treatment more than sham treatment was the group with a basal pressure greater than 40 mmHg. 

To assess whether sphincterotomy is an effective treatment for abdominal pain in patients with biliary type 3 SOD, the EPISOD trial was performed as a sham-controlled randomized trial that included 214 patients [16]. Patients included in the study were 18–65 years old with significant post-cholecystectomy biliary type pain and without evidence of prior pancreatitis or prior intervention on the biliary and/or pancreatic sphincter. Patients were randomized in a 2:1 ratio to sphincterotomy or underwent a sham procedure. Pain level and psychological parameters were assessed by several questionnaires. The primary outcome measure was the recurrent abdominal pain intensity and disability scale (RAPID) [8,10,16,37]. This study provided strong evidence that sphincterotomy was not more effective than the sham procedure in patients with type 3 SOD at short and longer-term follow-up at up to 5 years. The study recommended that sphincterotomy should not be performed in patients with biliary type 3 SOD [16,18].

### 5.2. Pharmacologic Treatment

The role of pharmacologic therapy for patients with SOD is not well explored [10,14,37,49]. According to the literature, several studies with different treatments have shown varying efficacies in patients with SOD, but all had a low level of evidence [10,14,30,43,49,50,51,52,53,54,55,56,57,58]. Moreover, from the literature, it is unclear whether SOD was investigated in pre-cholecystectomy or post-cholecystectomy patients, in addition to the small number of included patients. Treatment options are summarized in Table 4.

Calcium-channel blockers (nifedipine, nicardipine) are known smooth muscle relaxants that potentially could decrease the basal pressure of the SO [53,57]. Nifedipine has been explored in the SOD treatment in the randomized controlled trial [53] including 28 patients with elevated SO basal pressure without abnormal phasic wave contractions on SOM. Patients were treated with nifedipine or placebo over 12 weeks, with a subsequent cross-over. To monitor the level of biliary pain, patients kept diaries of pain levels and visits to the emergency department were registered. Compared with patients in the placebo group, patients in the nifedipine group had a significantly lower number of pain episodes and emergency room visits, and a decreased use of analgesics. Among all patients, 21 reported an improved pain level. Moreover, no significant differences in the tolerated nifedipine dose between groups were reported. However, patients with improved pain levels had predominantly antegrade propagation of phasic contractions, whereas patients who maintained their pain level had predominantly retrograde contractions. Thus, nifedipine decreased basal and phasic pressures, but did not have an effect on the sequence of phasic contractions [53]. 

These results suggested that nifedipine treatment is effective. However, the effect is rather subjective, since the results are dependent on perception and tolerance of the intensity of pain.

A short-term double-blind cross-over study [57] followed 13 patients with type 2 SOD after cholecystectomy for 16 weeks. All patients kept logs of pain and their need for analgesics or antispasmodics. They received nifedipine or placebo treatment with a cross-over at 8 weeks. A significant reduction in days with pain and use of analgesics consumed for patients receiving nifedipine treatment was reported as compared to patients receiving placebo [57]. After a study period of 16 weeks, the patients had the opportunity to continue the medication. At a median of 22 months of follow-up, 8 patients still received medication satisfactorily. 

Thus, nifedipine seemed to reduce pain effectively and safely in patients. Explorations of the underlying mechanisms indicated that calcium blockers significantly (more than 50%) inhibited the acetylcholine-induced and potassium chloride (KCl)-induced SO contractions in a dose-dependent manner [61]. Therefore, the smooth muscle selective calcium channel antagonists could be potent inhibitors of SO contractions. They may be potential treatment for type 3 SOD, which remains for future studies to explore. 

The effect of glyceryl trinitrate (GTN) on the SO motility and baseline pressure was investigated in a small pre-clinical study [54] including patients that underwent ERCP and SOM. Patients received sublingual GTN with manometry performed before and after, and were compared to controls. There was no difference in baseline SO pressure between these groups, but the group that received GTN had a significant decrease in the amplitude of contractions, with no difference in frequency of contractions. These results indicated that GTN treatment does not influence the sphincter of Oddi motility; however, the sphincter of Oddi muscle is relaxed very effectively. It can be concluded that GTN could be useful for the treatment of biliary colic and as a premedication before endoscopic examinations [54]. The clinical effects of GTN in the treatment of SOD were not explored. 

In a prospective study including 59 patients with biliary SOD (14% type 1, 51% type 2, 35% type 3), the outcome of biliary SOD managed without SOM was assessed [43]. All patients with a dilated common bile duct were offered ERCP and sphincterotomy, whereas all others were offered pharmacological treatment alone. Pharmacological treatments included a combination of nitrates, tricyclic antidepressants (amitriptyline) and/or analgesics, and patients were followed for a median of 15 months. At follow-up, 15.3% of patients reported complete symptom resolution, 59.3% improvement, 22% unchanged symptoms, and 3.4% deterioration. In total, 14/23 patients (61%) undergoing ERCP/sphincterotomy reported initial symptom resolution/improvement as a result of the ERCP interventions. In total, 21/59 (35.6%) patients experienced symptom resolution or improvement on low-dose tricyclic antidepressants (mainly amitryptiline: in two cases in combination with GTN spray and in two other cases in combination with tramadol), 3/59 (5%) on GTN spray, 2/59 on buscopan, 2/59 on morphine, and 1 each on nifedipine, low-dose citalopram, low-dose venlafaxine, diclofenac, paracetamol, tramadol, gabapentin (in combination with tramadol), and low-fat diet alone. Although 7/21 patients who were receiving opiates at baseline were weaned off these medications, another 6 patients were started on opiates (mainly tramadol) during the follow-up period. Thus, it was concluded that patients with biliary SOD may be managed with a combination of sphincterotomy, performed in those with dilated common bile duct, and pharmacological treatment [43]. However, larger randomized controlled trials are needed.

The injection of botulinum toxin (Botox) into SO [10,44] also causes relaxation of SO and, therefore, a temporary improvement in biliary pain after cholecystectomy for about 4 months. At the clinical level, the injection of botulinum toxin was beneficial in 55% of patients with type 3 SOD in two uncontrolled clinical series [58,60].

Duloxetine, a serotonin norepinephrine reuptake inhibitor (SNRI), was explored in an open-label, single-center, single-arm, 12-week trial [59] including 20 patients with suspected SOD. The primary outcome measure was a Patient Global Impression of Change (PGIC) scale. The effect of duloxetine (SNRI) in SOD patients is based on the finding that 5-HT and norepinephrine are involved in visceral pain pathways, and 5-HT has been shown to play a role in innervating the SO. Patients experienced a significant decrease in symptoms and it was therefore concluded that duloxetine could be further studied for the treatment of SOD. 

A randomized controlled trial investigated the use of ursodeoxycholic acid (ursofalk) in the treatment of 118 patients with pain after cholecystectomy and with the presence of bile microlithiasis, identified though EUS [52]. After treatment with ursofalk for a few months, the biliary-type abdominal pain was significantly improved or resolved, while patients in the control group showed no improvement in symptoms. Although the study excluded patients with SOD type 1 and 2, it identified a clinically reversible cause of post-cholecystectomy pain [52]. 

Based on the available evidence, we suggest a clinical decision-making flowchart in Figure 1.

Non-randomized studies indicate that response rates for medical therapies for SOD are similar to sphincterotomy [43,49] and suggest that medical treatment could be an alternative to endoscopic sphincterotomy for some patients. However, larger randomized controlled trials focusing on the medical treatment of SOD are lacking. Future randomized controlled trials including patients with SOD are needed in order to study the most appropriate pharmacological management approach. Such trials could include either placebo or sphincterotomy controls. Further, the diagnosis of SOD should rely on the more non-invasive modalities, such as those described above, in such trials.

## 6. Conclusions

Biliary SOD is classified into three types according to Milwaukee classification, based on clinical presentation as well as laboratory and/or imaging abnormalities. As a functional disorder, SOD is classified according to Rome IV criteria. Currently, patients with post-cholecystectomy pain and SOD are mostly diagnosed by the Rome IV criteria. The management of SOD most often involves non-pharmacologic treatment, and high-level evidence supports sphincterotomy in type 1 and 2 SOD. Emerging high-quality data show no effect of sphincterotomy for patients with type 3 SOD. However, ERCP with biliary sphincterotomy is associated with post-ERCP pancreatitis rates from 10% to 15%.

The role of pharmacologic treatment for patients with SOD is not well explored. The only studies with a higher level of evidence are available for calcium-channel blockers. The remaining literature on pharmacologic treatments have shown varying efficacies in patients with SOD, all without a sufficient level of evidence.

Thus, modern trends in healthcare require high-quality preventive protocols with non-invasive methods in prevalent diseases. Therefore, there is an unmet need for randomized controlled clinical studies evaluating non-invasive diagnostics and interventions for the treatment of SOD.

## Figures and Tables

**Figure 1 jcm-12-04802-f001:**
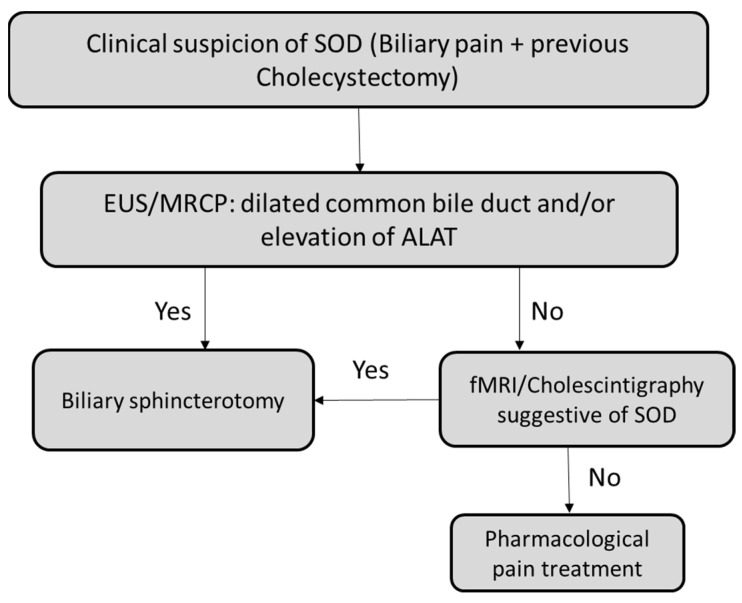
Decision-making in Sphincter of Oddi syndrome.

**Table 1 jcm-12-04802-t001:** Milwaukee classification of biliary sphincter of Oddi dysfunction [13].

**Type 1**	Biliary pain and all 3 of the following:
	-Serum aminotransferases: elevation of serum-aminotransferases above 2 times the upper limit on 2 or more occasions.
	-CBD* dilation: above 10 mm on US ** or above 12 mm on ERCP ***.
	-Delayed drainage of contrast from the CBD * on ERCP ***.
**Type 2**	Biliary pain and 1 or 2 out of the 3 above criteria.
**Type 3**	Biliary pain.

* CBD = common bile duct, ** US = ultrasound, *** ERCP = endoscopic retrograde cholangiopancreatography.

**Table 2 jcm-12-04802-t002:** Rome IV criteria for biliary SOD [31,34,35].

Biliary SOD
Biliary pain. Absence of bile duct stones or other structural abnormalities. Elevated liver enzymes or dilated bile duct (but not both). Supportive criteria: Normal amylase/lipase. Abnormal Sphincter of Oddi manometry. Abnormal hepatobiliary scintigraphy.

**Table 3 jcm-12-04802-t003:** Diagnostic tools for biliary SOD.

Diagnostic Tools	Short Description of the Method	Strengths	Limitations
Sphincter of Oddi manometry [2,5,36,37]	Performed during ERCP *. A catheter is inserted into the bile duct.	Can directly assess SO ** motor activity.	Risk of post-ERCP * pancreatitis. Rare availability due to need for highly trained expertise.
Functional MRI *** [37,46]	Measurement of biliary contrast agent excretion to the duodenum.	Non-invasive.	Cannot be used on patients with metal devices, claustrophobia or intolerance of contrast.
Optical Coherence Tomography [37,45]	A probe is inserted into the CBD **** through an ERCP * catheter. Low-power infra-red light allows for visualization of the SO ** microstructure, which is thickened in patients with SOD.	High-resolution, real-time imaging.	Risk of post-ERCP * pancreatitis.
Functional Lumen Imaging Probe technique [37]	Performed during ERCP*	Analyzes the sphincter profile and motility patterns.	Risk of post-ERCP * pancreatitis. Need for highly trained expertise.
Hepatobiliary scintigraphy (cholescintigraphy) [1,13,23,38,44,47]	Measurement of time–activity curve for excretion of a radio nucleotide (injected IV) via the hepatobiliary system.	Assesses the rate of bile flow into the duodenum.	Quite challenging to interpret due to lack of consensus on its diagnostic use.

* ERCP = endoscopic retrograde cholangiopancreatography, ** SO = Sphincter of Oddi, *** MRI = Magnetic Resonance Imaging, ****CBD = Common Bile Duct.

**Table 4 jcm-12-04802-t004:** Treatment options for biliary SOD [10,14,30,43,49,50,51,52,53,54,55,56,57,58].

Treatment: Non-P */P **	SOD Type/Level of Evidence ***	Short Description of the Method	Strengths	Limitations
Non-P *: Biliary sphincterotomy [2,17]	Biliary-type pain/biliary obstruction /Level 2	Performed during ERCP.	Most definite treatment option.	Risk of post-ERCP pancreatitis. Need for highly trained expertise. No clinical effect for type 3 SOD.
Non-P *: Biliary sphincterotomy [16]	SOD type 3 /Level 2			
P **: Calcium-channel blockers [53,57]	SOD type 2 /Level 2	Inhibit SO contractions and decrease basal pressure of the SO.	Non-invasive	Only short-term effects were explored.
P **: Serotonin norepinephrine reuptake inhibitors (SNRIs) [59]	SOD type 3 /Level 4	5-HT-receptor-mediated pain relief.	Non-invasive	Usual precausions (allergies, intolerance, side effects, etc.).
P **: Tricyclic antidepressants (amitriptyline) [43]	SOD type 1, 2 and 3 /Level 5	Relax the SO.	Non-invasive	Only short-term effects were explored.
P **: Glyceryl trinitrate [43,54]	SOD type 1, 2 and 3 /Level 5	Relaxes the SO musculature.	Non-invasive	Only short-term effects were explored.
P **: Injection of botulinum toxin into SO [58,60]	SOD type 3 /Level 4	Decreases basal pressure of the SO.	Temporary pain relief (about 4 months)	Invasive method.
P **: Ursodeoxycholic acid (Ursofalk) [52]	SOD type 3 /Level 4	Dissolves biliary crystals that can cause a biliary pain.	Non-invasive	Usual precausions (allergies, intolerance, side effects, etc.).

Non-P * = non-pharmacologic treatment, P ** = pharmacologic treatment, *** The Center for Evidence-Based Medicine.

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
