# Peer review of "Dysfunction of Biliary Sphincter of Oddi—Clinical, Diagnostic and Treatment Challenges"

_jcm, 2023, doi:10.3390/jcm12144802_

Round 1

Reviewer 1 Report

Is there any consensus, publication, or other work to help understand/differentiate pre-cholecystectomy and post-cholecystectomy SOD? Is pre-cholecystectomy SOD a possible entity? For example, is there a  risk/rate of suspected symptomatic gallstone disease that is actually SOD? Are you focusing solely on post-cholecystectomy SOD throughout the paper? Be clear when it is post-cholecystectomy SOD, or pre-cholecystectomy, if that is possible, especially when assessing/interpreting the prior literature. 

I find your emphasis of needed understanding of HIDA vs EUS or Endoscopy in SOD, or prevention of EUS or Endoscopy in SOD, refreshing, because we need less invasive diagnostics and interventions in prevalent diseases, like gallstone disease and potentially SOD. Potentially tie this point to your call for future RCTs (see below). Please feel free to reference, if applicable: 

Peck GL, Hudson SV, Roy JA, Gracias VH, Strom BL. Use of a New Prevention Model in Acute Care Surgery: A Population Approach to Preventing Emergency Surgical Morbidity and Mortality. Ann Surg Open. 2022 Sep;3(3):e188. doi: 10.1097/as9.0000000000000188. PMID: 35990734; PMCID: PMC9390954.

The paragraph that introduces pharmacologic treatments would benefit from reiterating / specifying whether these data/studies are in pre-cholecystectomy or post-cholecystectomy patients. If that is not known, state that. This will be important context for assessing the current literature and conducting future studies of SOD. 

In the same text, can you clarify the implications of anterograde and retrograde phasic contractions with respect to mechanisms of SOD?

Can you develop sub-sections under "treatment options" section, possibly of non-pharmacologic treatments and pharmacologic treatments of SOD? Also, can you make that distinction in Table 4? 

Also, maybe a new paragraph in the conclusion starting with "The role for pharmacolgic treatment . . . "

Small point, I am sure there are some limitation to SNRI and UDCA? I would include in the table for completeness. 

Would HIDA or functional MRI be useful in the RCTs that you propose are necessary of pharmacologic treatments? Would you advocate for what modality in the proposed RCTs?

At this stage of the research paradigm, is the need for randomized controlled clinical studies evaluating pharmacological treatment of SOD the next step, the only step, or are there other steps including large observational research studying pharmacologics indicated? Perhaps adding that in as another possible next step would advance the call for more research on SOD, more preliminary data that justifies an RCT, and given how expensive and invasive an RCT would be.

Author Response

Reviewer 1:

Thank you very much for the relevant and useful comments. We hope you will find our response satisfying. Attached you will find the revised manuscript with the changes highlighted in red.

  1. Is there any consensus, publication, or other work to help understand/differentiate pre-cholecystectomy and post-cholecystectomy SOD? Is pre-cholecystectomy SOD a possible entity? For example, is there a risk/rate of suspected symptomatic gallstone disease that is actually SOD? Are you focusing solely on post-cholecystectomy SOD throughout the paper? Be clear when it is post-cholecystectomy SOD, or pre-cholecystectomy, if that is possible, especially when assessing/interpreting the prior literature.

Answer: Dear Reviewer, thank you for your comments.

We are mainly focusing on post-cholecystectomy SOD. To our knowledge, the literature used in present review is also focusing on post-cholecystectomy SOD. However, the existing literature on the topic, as well as original research are scarce, therefore in general it is quite hard to separate the pre-cholecystectomy SOD and post-cholecystectomy SOD, since the exact mechanism of SOD is still uncertain.

We added: Moreover, from the literature it is unclear whether SOD was investigated in pre-cholecystectomy or post-cholecystectomy patients, line 228-229.

  1. I find your emphasis of needed understanding of HIDA vs EUS or Endoscopy in SOD, or prevention of EUS or Endoscopy in SOD, refreshing, because we need less invasive diagnostics and interventions in prevalent diseases, like gallstone disease and potentially SOD. Potentially tie this point to your call for future RCTs (see below). Please feel free to reference, if applicable:

Peck GL, Hudson SV, Roy JA, Gracias VH, Strom BL. Use of a New Prevention Model in Acute Care Surgery: A Population Approach to Preventing Emergency Surgical Morbidity and Mortality. Ann Surg Open. 2022 Sep;3(3):e188. doi: 10.1097/as9.0000000000000188. PMID: 35990734; PMCID: PMC9390954.

Answer: Thank you! Very interesting reference and suggestion, it is added (ref. 20, lines 42-47 and 318-321).

  1. The paragraph that introduces pharmacologic treatments would benefit from reiterating / specifying whether these data/studies are in pre-cholecystectomy or post-cholecystectomy patients. If that is not known, state that. This will be important context for assessing the current literature and conducting future studies of SOD.

Answer:  We added more information accordingly, line 228-229.

  1. In the same text, can you clarify the implications of anterograde and retrograde phasic contractions with respect to mechanisms of SOD?

Answer: We added a new paragraph (Lines 64-95).

  1. Can you develop sub-sections under "treatment options" section, possibly of non-pharmacologic treatments and pharmacologic treatments of SOD? Also, can you make that distinction in Table 4?

Answer: Thank you, sub-headings are now added, and distinction is made in Table 4.

  1. Also, maybe a new paragraph in the conclusion starting with "The role for pharmacolgic treatment . . . "

Answer: Now added (line 338).

  1. Small point, I am sure there are some limitation to SNRI and UDCA? I would include in the table for completeness.

Answer: We agree, relevant info is now added in table 4.

  1. Would HIDA or functional MRI be useful in the RCTs that you propose are necessary of pharmacologic treatments? Would you advocate for what modality in the proposed RCTs?

Answer: Yes, absolutely. See comments lines 342-45.

  1. At this stage of the research paradigm, is the need for randomized controlled clinical studies evaluating pharmacological treatment of SOD the next step, the only step, or are there other steps including large observational research studying pharmacologics indicated? Perhaps adding that in as another possible next step would advance the call for more research on SOD, more preliminary data that justifies an RCT, and given how expensive and invasive an RCT would be.

Answer: Thank you for your comment. We have added further comments (lines 342-45). Regarding larger observational studies from databases or other data sources as such, we do not believe that those are feasible since SOD is a rare condition.

Reviewer 2 Report

The authors describe SOD, paying attention to symptoms, diagnosis and treatment.

I have a few comments about the article:

1.                  What criteria were used to include and exclude articles for review? I understand that this is a literature review, but it seems interesting to present what method of selecting articles the authors used.

2.                  Line 29: Please explain the term "inflammatory fibrosis"

3.                  In Table 3: "Findings may be influenced by a number of factors". Please give a detailed explanation.

4.                  In Table 3: „Usual limitations of MRI scans”. Please give a detailed explanation.

5.                  Line 181-194: It is worth noting that 12 patients participated in the study.

6.                  On what basis the authors assessed the "Level of evidence" in Table 4. No references.

7.                  In Table 4 SNRIs have no limitations. Shouldn't information be added about the small number of studies and the small group of patients included in the study?

8.                  In line 230, Ca-blockers have a subjective effect, and in line 240, Ca-blockers are effective. Please coherently summarize the effect of Ca-blockers

9.                  According to reference 42, what is the best therapeutic option. Please summarize.

10.               Botox provides temporary improvement. Has it been studied how long the effect lasts?

11.               It will be important for readers how to proceed in everyday clinical practice with patients. Could the authors show this in a diagram/figure?

Author Response

Reviewer 2:

The authors describe SOD, paying attention to symptoms, diagnosis and treatment.

Thank you very much for the relevant and useful comments. We hope that you will find our response satisfying. Attached you will find revised manuscript with changes (from both reviewers) highlighted in red.

I have a few comments about the article:

  1. What criteria were used to include and exclude articles for review? I understand that this is a literature review, but it seems interesting to present what method of selecting articles the authors used.

Answer: Dear Reviewer, thank you for your comments.

In our work we are focusing mainly on post-cholecystectomy. However, the existing literature on the topic, as well as original research are scarce. In order to get an overview on current knowledge on biliary SOD we searched on pubmed database and screened the reference lists from the studies we were aware of.

  1. Line 29: Please explain the term "inflammatory fibrosis"

Answer: The term "inflammatory fibrosis" means the following: due to chronic inflammation in the bile duct, a fibrosis or scarring occurs by the accumulation of excess extracellular matrix components. That can lead to narrowing of the diameter of the bile duct or its eventual stenosis. The term was changed to “fibrosis due to chronic inflammation of bile duct” in the manuscript.

  1. In Table 3: "Findings may be influenced by a number of factors". Please give a detailed explanation.

Answer: We rephrased in the text (lines 134-138) and in Table 3.

  1. In Table 3: „Usual limitations of MRI scans”. Please give a detailed explanation.

Answer: This info is now added.

  1. Line 181-194: It is worth noting that 12 patients participated in the study.

Answer: Of course, we added “small” study.

  1. On what basis the authors assessed the "Level of evidence" in Table 4. No references.

Answer: The "Level of evidence" in Table 4 is accessed based on the scheme from The Center for Evidence-Based Medicine (the footnote in Table 4).

  1. In Table 4 SNRIs have no limitations. Shouldn't information be added about the small number of studies and the small group of patients included in the study?

Answer: We agree, we added this info and we added …, in addition to the small number of included patients, line 229-230.

  1. In line 230, Ca-blockers have a subjective effect, and in line 240, Ca-blockers are effective. Please coherently summarize the effect of Ca-blockers

Answer: Thank you, we have rephrased accordingly (line 251-253).

  1. According to reference 42, what is the best therapeutic option. Please summarize.

Answer: Now it is a reference 43, since reference 20 was added. It is updated (line 294-297).

  1. Botox provides temporary improvement. Has it been studied how long the effect lasts?

Answer: The effect from Botox lasts for about 4 month. This info is now added to the Table 4 and line 299-300.

  1. It will be important for readers how to proceed in everyday clinical practice with patients. Could the authors show this in a diagram/figure?

Answer: We agree, we have added a new diagram.
